# Using Reinforcement Learning to Develop a Novel Gait for a Bio-Robotic California Sea Lion

**DOI:** 10.3390/biomimetics9090522

**Published:** 2024-08-30

**Authors:** Anthony Drago, Shraman Kadapa, Nicholas Marcouiller, Harry G. Kwatny, James L. Tangorra

**Affiliations:** Laboratory for Biological Systems Analysis, Department of Mechanical Engineering and Mechanics, Drexel University, Philadelphia, PA 19104, USA; sk3496@drexel.edu (S.K.); nm875@drexel.edu (N.M.); hgk22@drexel.edu (H.G.K.); jlt66@drexel.edu (J.L.T.)

**Keywords:** bio-robotics, gait development, reinforcement learning, sea lion, bio-memetic propulsion

## Abstract

While researchers have made notable progress in bio-inspired swimming robot development, a persistent challenge lies in creating propulsive gaits tailored to these robotic systems. The California sea lion achieves its robust swimming abilities through a careful coordination of foreflippers and body segments. In this paper, reinforcement learning (RL) was used to develop a novel sea lion foreflipper gait for a bio-robotic swimmer using a numerically modelled computational representation of the robot. This model integration enabled reinforcement learning to develop desired swimming gaits in the challenging underwater domain. The novel RL gait outperformed the characteristic sea lion foreflipper gait in the simulated underwater domain. When applied to the real-world robot, the RL constructed novel gait performed as well as or better than the characteristic sea lion gait in many factors. This work shows the potential for using complimentary bio-robotic and numerical models with reinforcement learning to enable the development of effective gaits and maneuvers for underwater swimming vehicles.

## 1. Introduction

In recent years, there has been significant interest in developing bio-inspired swimming vehicles that emulate the propulsive methods of biological systems. Biological swimmers utilize their bodies and propulsors to achieve remarkable maneuverability and agility, inspiring engineers to model these systems for improving the performance of underwater vehicles [1,2]. Existing bio-inspired swimming robots include fish robots that make use of multiple fins to produce propulsive forces and maneuvers [3,4,5,6], sea turtles with soft actuator-driven flippers [7], and sea snakes capable of contorting to navigate tight areas [8]. This work focuses on the California sea lion, which is an excellent model for bio-inspired robotic swimming systems due to its exceptional maneuverability and agility, especially in high-energy flow environments [9]. Its unique propulsion method, which relies heavily on the coordinated movement of its foreflippers, allows it to achieve impressive swimming performance [10]. The coordination of these foreflippers for a propulsive stroke is understood to follow a consistent characteristic gait [10]. This characteristic stroke actuates the flipper at its base with a three-degrees-of-freedom shoulder joint and is deployed in tandem with body motions to achieve the agile swimming displayed by the sea lion [9]. While the sea lion’s natural gait is effective for its purposes, this work explores if this gait is ideal for a bio-robotic model of the sea lion if the characteristic biological stroke can be used to produce effective swimming with just its propulsors.

Reinforcement learning (RL) offers a promising approach to address the challenge of coordinating the propulsive elements of a bio-robotic system inspired by the California sea lion and potentially produce new novel effective foreflipper strokes. RL has proven effective in terrestrial applications by transferring quadruped animal walking gaits onto robotic quadrupeds [11] and teaching humanoid bipedal robots how to walk effectively [12]. Additionally, in an underwater context, RL has been successfully deployed to train a beaver-like swimmer [13], underwater armed manipulators [14], and a simple two-degrees-of-freedom fin-based swimming system [15]. High-fidelity models are essential for RL due to the substantial number of trials required for effective learning [16,17]. Training directly on a numerical model reduces the risk of damage, shortens training time, and enhances state space exploration, potentially improving gait outcomes. However, developing a high-fidelity underwater bio-robotic model is computationally costly due to fluidic complexities and lengthy simulation times. Any simulation of underwater swimming robots used for RL must produce accurate results while operating at relevant time scales.

To use reinforcement learning (RL) to coordinate the sea lion propulsors effectively for bio-robotic swimming, the characteristic flipper kinematics used by the animal during swimming must be understood, and a high-fidelity model of the robotic system must be developed. In this paper, reinforcement learning was applied to a high-fidelity numerical model of a multi-body underwater bio-robotic system to develop new straight swimming propulsive strokes (Figure 1). These learned strokes were then implemented on the bio-robotic system to verify its effectiveness for straight swimming, with performance comparisons made against the numerical model. To compute hydrodynamic coefficients for the simulation model, we leveraged computational fluid dynamics (CFD) and strip theory, achieving close approximations without the need for expensive tow-tank experimentation. This methodology offers a rapid and computationally effective alternative to traditional experimental techniques, significantly advancing the potential of bio-robotic propulsion systems.

The objective of this work was to evaluate the effectiveness of applying reinforcement learning to modify a characteristic California sea lion propulsive stroke to produce new straight swimming gaits and evaluate the novel gaits’ performance on a swimming bio-robotic sea lion (Figure 1). The process that was followed included (1) an analysis of the sea lion during natural swimming, (2) the development and validation of bio-robotic and numerical models of a California sea lion, (3) the application of reinforcement learning to train the kinematics of the foreflippers in simulation to produce desired swimming performance, and (4) a comparison of performance of the observed characteristic biological sea lion swimming gait and the learned gait in simulation and on the bio-robotic platform (Figure 1). 

## 2. Materials and Methods

### 2.1. The Sea Lion Foreflipper Stroke Model

Videos of the California sea lion (Zalophus californianus) were analyzed to identify the characteristic gaits and body motions used during natural swimming. Unmarked and non-research sea lions were observed and filmed at the Smithsonian Zoological Park through a sizable underwater glass viewing window large enough to capture, at minimum, 3–4 sea lion body lengths [9,10]. Using a stationary camera (GC-PX100BU, JVC, Japan), videos were recorded as the sea lions passively swam (Figure 2). Crucial factors were identified, including a manipulatable head/cervical section, flexible pelvic section, hind flippers for control, and foreflippers as primary propulsive devices. The kinematics of the characteristic sea lion stroke have been well described in previous works [10]. Additional videos were recorded from several different angles of multiple sea lions freely swimming in a zoo setting (Figure 2). The video footage was analyzed to determine the foreflipper motion at the base of the foreflipper (Figure 2). The resulting kinematics were in alignment with previous works. These tracked kinematics were used as the basis for the development of a generalized model for the kinematics of the sea lion foreflippers.

The sea lion propulsive stroke cycle is divided into three distinct phases: recovery, power, and paddle. During the recovery phase, the foreflipper aligns with the flow direction through yaw and roll motions, extending laterally for low drag, and setting up for the propulsion direction (Figure 2 and Figure 3). The subsequent power phase involves pitch rotation, pulling the flipper towards and beneath the body medially, with decreased roll orienting the leading edge towards the motion direction (Figure 2 and Figure 3). This is followed by the paddle phase, where the flippers yaw and roll inward, aligning the flipper face with the motion direction and concluding with the flippers in a streamlined position beside the body direction (Figure 2 and Figure 3).

A generalized and parameterized model of a sea lion’s foreflipper stroke was developed. A piecewise cubic Hermite interpolating polynomial (PCHIP) spline was fit by hand to generalized kinematics obtained from a combination of the sea lion videos and previous work on the sea lion stroke [10] (Figure 4). PCHIP splines offer bounded magnitudes between control points and support double differentiation, ensuring a smooth trajectory (Figure 4). By adjusting the control points of the spline, the flipper trajectory can produce a range of motions, from accurately mimicking the sea lion’s characteristic propulsive stroke to entirely random movements resulting in no forward propulsion. More specific alterations for the purpose of reinforcement learning will be discussed in Section 2.4.

### 2.2. The Bio-Robotic Sealion

A bio-robotic system known as the Stroke Experimentation and Maneuver Optimizing Underwater Robot (SEAMOUR) was developed to model the swimming and maneuvering of the California sea lion. The important biological features for swimming and maneuvering have been identified by experts+. The bio-robotic system models five important segments of the sea lion for swimming and maneuvering: the head/neck section, the main body, the foreflippers, the pelvic section, and the hind flippers (Figure 5). The position and scale of these sections were based off the sea lion animal. In addition, like the animal, SEAMOUR is laterally symmetric down the centerline of the body but is about half the total length at 1 m. SEAMOUR has access to 14 degrees of freedom (DoF), which are driven by waterproof servo motors (XPERT RC WR-7701, XPERT USA, Bellevue, WA, USA). The head and pelvic sections both utilize a two-axis gimbal system for yaw and pitch motion that enables them to transform 60° in all directions (Figure 3). The pelvic section houses four motors, two for each flipper, that give the hind flippers their ability to roll and yaw. Each foreflipper is driven by three servo motors, giving the flipper the ability to roll, pitch, and yaw (Figure 6). These waterproof motors surpassed both the speed and estimated torque requirements for the foreflipper gaits to be executed. To enhance the foreflipper assembly’s rigidity, both the pitch and roll motors are supported on-axis with bearings. However, due to the trajectories of certain gaits and space constraints, the yaw axis motor could not be similarly supported (Figure 6).

The foreflipper is made with an ABS 3D printed (F120, Stratasys, Eden, MN, USA) grid support structure that is cast into an uncambered airfoil shape using a two-part silicone (Smooth-on, Macungie, PA, USA) with a shore hardness of 00-30. While the foreflipper shape was simplified, careful consideration was taken to preserve the bending location and the amount of tip displacement during swimming (Figure 7). Using Solidworks 2022 CAD software (Solidworks 2022, Waltham, MA, USA), finite element analysis (FEA) was done to simulate the bending of the foreflipper when subjected to an estimated peak fluidic loading during a characteristic propulsive stroke. Since the rigidity of the flipper comes from the ABS support structure, the simulation was run without the properties of the silicone. A 5N non-uniform distributed load was applied along the length of the ABS support structure, and the displacement of the tip was measured (Figure 7).

The body of the robotic platform is a streamlined ellipsoid shape with the power and control components housed in a waterproof box located in the interior section (Figure 6). SEAMOUR is operated with a modified Raspberry Pi 4 (RPI), equipped with an extension antenna designed to breach the water’s surface, enabling untethered remote control (Figure 6). The system is powered with a single 2250 mAh 3-cell LiPo battery (MaxAmps, Spokane, WA, USA), as well as a 5 V voltage regulator for the RPI (Pololu, D36V28F5 Las Vegas, NV, USA) and an 8.4 V 15 amp voltage regulator for the servo motors (Pololu, D24V150F6, Las Vegas, NV, USA) (Figure 6). To control the servo motors, a daughter board (PN 2327, Adafruit, Brooklyn, NY, USA) utilizing pulse-width modulation (PWM) was added to the RPI (Figure 6). Adding mass and extruded polystyrene foam to various locations throughout the body is employed not only to trim the system’s pitch and roll but also to modify the desired center of mass and center of buoyancy as well as to achieve neutral buoyancy. The center of buoyancy was always located above the center of gravity to provide pitch and roll stability, as is often done for bio-robotic swimmers [18].

### 2.3. The Numerical Model of the Bio-Robotic Sea Lion

A numerical model of SEAMOUR was developed to test swimming gaits and body motions and served as a reinforcement learning training environment in an underwater domain. Created using Simscape (MATLAB 2023A and Simscape Toolbox Release 2022a, The MathWorks, Inc., Natick, MA, USA), this model is a true-to-scale representation of SEAMOUR, providing a detailed simulation of its mechanical components. It allows for realistic visualization and simulation of the robot’s 6 DoF body movements (Figure 5C). Within the Simscape model, the interaction of the multi-body system with the environment can also be modeled. The model accounted for inertial forces and the hydrostatic and hydrodynamic forces experienced by the bio-robotic system underwater. In this case, the flow was assumed to be incompressible and inviscid. Each part of SEAMOUR was modeled in SOLIDWORKS 2022 and added to the model. The weights of the individual components, head, body, pelvic section, hind flippers, and foreflippers, were measured and added to the Simscape model. Joint torques at each connection point are automatically calculated in the Simscape environment, and the passive bending in the foreflippers was modeled through a spring-mass damper connecting two rigid flat plates (Figure 7).

To model this complex bio-robotic system, several assumptions were made: the model was neutrally buoyant and fully submerged in a fluid that is incompressible and has no skin friction [19]. The model assumed symmetry in all three axes. The main body, including the head and pelvic section, was considered a prolate spheroid (Figure 8). The foreflippers and hind flippers were considered as rectangular flat plates. SEAMOUR was designed as an open system that allows water ingress. The water volume within the robot was estimated by subtracting the combined volume of its internal components from the total robot volume, incorporating it as additional mass. The center of mass for the water and individual sections was also included in the model.

#### 2.3.1. Hydrostatic Forces (Gravity and Buoyancy)

Hydrostatic forces, including both gravity and buoyancy, were considered for each core component of the model. The center of buoyancy coincided with the center of gravity for the flippers, head, and pelvic section. For the main body, the center of gravity was positioned directly below the center of buoyancy with a slight offset to enhance stability along the roll and pitch axes and reflect the actual center of mass and center of buoyancy of SEAMOUR.

#### 2.3.2. Hydrodynamic Forces (Drag and Added Mass)

To model the fluid forces, hydrodynamic forces such as drag and added mass forces were calculated and applied. In the x-direction, drag and added mass forces were applied to the center of mass (CoM) of the main body. Due to the dynamically moving head and pelvic section in this multi-body model, the primary body was divided into eight sections down the roll axis of the body (Figure 8D). Drag and added mass forces, proportional to each section’s surface area, were computed and applied at the centers of each section (Equations (1) and (7)–(9)). For the flippers, these hydrodynamic forces were integrated along the length of the flipper and applied directly to the center of mass of the flippers. To compute the drag coefficient in the x-direction, a simplified three-dimensional computational fluid dynamic simulation was developed using COMSOL Multiphysics (COMSOL, Inc., Boston, MA, USA) (Figure 8A–C). A 3D model of SEAMOUR’s shape was imported directly into COMSOL and positioned in the center of a large water tank measuring 14 m × 8 m × 6 m. The k-ε turbulence model with Reynolds-averaged Navier-Stokes (RANS) formulation was employed to simulate the fluid flow. A user-controlled coarse mesh calibrated for fluid dynamics was used (Figure 8A). The front wall of the tank served as the inlet, while the back wall acted as the outlet, maintaining a uniform flow regime. The integration of the total stress due to pressure force along the body produced the drag force in the x-direction. Using Equation (1), the x-direction drag coefficient was computed.
(1)$$C_{d}=\frac{2\;F_{D}}{\rho v^{2}A}$$
where $C_{d}$ is the drag coefficient, ρ is the density of fluid, v is the velocity of the body, $F_{D}$ is the drag force, and A is the cross-sectional area. Each of the eight sections of the main body were positioned in the center of a large tank in the CFD model, similar to the main body in the x-direction, and force coefficients were computed in the y- and z-directions (Figure 8D). For the foreflippers and hind flippers, the largest face was treated as a flat plate with a drag coefficient of 1.28, while the coefficient for the other long, thin face was taken as a symmetric air foil with a drag coefficient of 0.08. For added mass coefficients, the main body was considered as a prolate spheroid. Using the eccentricity and other geometric constants, the added mass coefficients in the x-, y-, and z-directions were computed [18], as shown in Equations (2)–(6).
(2)$$e=\sqrt{1-{\left(b/a\right)}^{2}}$$
(3)$$\alpha_{0}=\frac{2\left(1-e^{2}\right)}{e^{3}}\left(\frac{1}{2}\ln\frac{1+e}{1-e}-e\right)$$
(4)$$\beta_{0}=\frac{1}{e^{2}}-\frac{1-e^{2}}{2e^{3}}\ln\frac{1+e}{1-e}$$
(5)$$m=\frac{4}{3}\pi \rho ab^{2}$$
where *e* is the eccentricity of the prolate spheroid, *a* is the semi-major axis and *b* is the semi-minor axis, m is the mass, and $\alpha_{0}$ and $\beta_{0}$ are constants. The added mass coefficients in the y- and z-directions were scaled based on the surface area of each eighth section. These coefficients were then used to compute the added mass forces [18] using Equations (7)–(9). For the foreflippers and hind flippers, the added mass coefficients were computed using strip theory [19] in Equation (10). Equations (7)–(9) were then used to apply added mass forces to the flippers.
(6)$$X_{\dot{u}}=-\frac{\alpha_{0}}{2-\alpha_{0}}m;\;Z_{\dot{w}}=Y_{\dot{v}}=-\frac{\beta_{0}}{2-\beta_{0}}m$$
(7)$$X_{u}=X_{\dot{u}}\dot{u}$$
(8)$$Y_{v}=Y_{\dot{v}}\dot{v}$$
(9)$$Z_{w}=Z_{\dot{w}}\dot{w}$$
where $X_{u},\;Y_{v}$, and $Z_{w}$ are the hydrodynamic added mass force, $X_{\dot{u}},\;Y_{\dot{v}}$, and $Z_{\dot{w}}$ are the added mass coefficients, and $\dot{u},\dot{v}$, and $\dot{w}$ are the linear accelerations in the x-, y-, and z- directions, respectively.
(10)$$m_{a}=\frac{\pi \rho }{4}kbc^{2}$$
where ρ is the density of fluid, *k* is the coefficient of additional mass, *b* is the span of the rectangular plate, and *c* is the chord of the rectangular plate.

A detailed description of the assembly, design choices, solver configuration, and validity of the numerical model will be presented in a subsequent publication.

### 2.4. Reinforcement Learning

A reinforcement learning agent was applied to learn a novel straight swimming gait. The reinforcement learning agent selected was a soft actor-critic (SAC), of which the algorithm and structure are described in Haarnoja et al. [20]. SAC agents provide many advantages for the purposes of learning effective robotic gaits. First, SAC can work with a continuous action space, which is necessary to properly manipulate the motor trajectories. They are also robust to noisy and uncertain environments, which occur frequently in real world environments and complicated multi-body simulations. Even though soft actor-critic (SAC) agents may require more effort to implement compared to other popular alternatives like proximal policy optimization (PPO), their effectiveness in handling continuous action spaces and ability to scale well with high-dimensional action spaces make them a desirable agent for this application. The structure of the critic and actor networks that train the agent are shown in Figure 9. Pilot studies were conducted to improve hyperparameter tuning of the SAC (Figure 9).

The SAC agent was trained in simulation to optimize a single set of foreflipper kinematics to be applied to the bio-robotic system. Each training episode involved executing one action and recording the resultant observation, with the action being a cyclic fore-flipper gait repeatable any number of times to assess swimming performance. Observations were logged at each episode’s end. The highest reward-producing kinematics from the last 10% of the training episodes were selected as the “learned gait”. Three full training cycles were conducted with different initialization seeds resulting in three unique “learned gaits”. Each of these gaits was evaluated against the performance of the characteristic sea lion stroke in simulation and on the bio-robotic system. The SAC agents output a set of repeatable motor kinematics representing three distinct gaits. The SAC agents were not uploaded into the Raspberry Pi that runs the bio-robotic system. The outputs of the SAC agents were collected on a separate desktop computer and pre-loaded onto the Raspberry Pi prior to experimentation. This method offers several advantages: It reduces the computational demand by using pre-learned motor kinematics without requiring the full SAC agent onboard the Raspberry Pi. Additionally, the lack of real-time position or velocity sensors makes a closed-loop controller unfeasible. The cyclic nature of the gait and the inability to control other body movements mean that the effects of consecutive strokes are cumulative, making a single set of kinematics sufficient. Lastly, selecting a single set of kinematics rather than continuously pulling actions from the SAC agent ensures more consistent gait outcomes, facilitating better performance evaluation against the biological model. The learning episode was twelve seconds long and consisted of a single learning step that produced four repeated stroke cycles.

The coordinates of the control points of the generalized flipper stroke model served as the action space for the reinforcement learning agent. Specifically, the action space consisted of the timings and magnitudes of multiple points that control the pitch, yaw, and roll flipper kinematics, totaling 13 individual actions (Figure 8A). The actions that change the magnitude and timings of the control points were bound to be within the operational range of the motors that actuate the flipper kinematics (Figure 10A). The control points of the model that are not directly altered in the action space serve several purposes, including maintaining the cyclic shape of the stroke, bounding motor velocities to within the operational range, and preparing the flipper to reset at the end of a stroke, or they are directly tied to the control points changed by the agent. The PCHIP spline fit to the control points ensured a smooth transition between points without overshooting the set motor ranges and maintained twice differentiability of the trajectory. The period of the characteristic stroke was doubled, and an additional control point was added to change the pause time between flipper strokes. The controlled flipper kinematics were symmetric between the left and right flipper. The agent lacked access to additional degrees of freedom in the bio-robotic system like the head, pelvis, and hind flippers to simplify the initial reinforcement learning task and increase the likelihood of achieving an effective straight swimming gait (Figure 10A).

For each learning episode, the body translation and translational velocities and the body angle and angular velocities along each axis were taken as the observation space for the reinforcement learning agent (Figure 10B). There were 12 total observations recorded at the beginning and end of each reinforcement learning episode. The learning episode was twelve seconds long and consisted of a single learning step that produced four repeated stroke cycles with pauses in between. Given this single-step framework, observations and actions were updated only once per episode, a feature that could adversely impact convergence (Figure 10B). To address this issue, multiple full training cycles were executed in simulation with different initialization seeds (Figure 10B).

The goal of the reinforcement learning was to encourage a straight swimming gait and discourage any out-of-plane motion. This task was facilitated by the following reward function:(11)$$r=x+\left|V_{x}\right|\cdot V_{x}-\left(\left|V_{y}\right|+\left|V_{z}\right|\right)-\sqrt{{\dot{\Phi }}^{2}{+\dot{\theta }}^{2}{+\dot{\psi }}^{2}}-\sqrt{\Phi^{2}{+\theta }^{2}{+\psi }^{2}}$$
where x was the total forward distance traveled at the end of the episode, and $V_{x}$, $V_{y}$, and $V_{z}$ were the mean velocities over the episode, with Vx being squared to further incentive forward motion. Φ, θ, and ψ were the three angles that represent the heading of the bio-robotic system at the end of the episode, and $\dot{\Phi }$, $\dot{\theta }$, and $\dot{\psi }$ were their respective mean angular velocities of the duration of the episode. All of these terms were equally weighted.

### 2.5. Experimental Setup and Data Collection

Experimental trials for SEAMOUR were conducted in a large diving well, which measures 9 × 9 × 4 m. The experiments took place along one wall of the diving pool. An L-shaped aluminum structure was constructed to serve as a docking station for the bio-robotic system, ensuring consistent initial positioning and orientation for each trial (Figure 11A). This structure included a magnet on its underside to attach to the robot’s core; this attachment could be disengaged from above the water. Testing was conducted at a consistent depth of 1.5 m (Figure 11A). At the start of each experiment, SEAMOUR was released from the dock, enabling unconstrained 3D swimming. A GoPro Hero 11 camera was used to record footage of each experiment at 24 frames per second at a 4K resolution. The camera was positioned at the center of the testing area on the opposite side of the pool, capturing a side view of the experimental trials from 8 m away. A fixed meter scale was used as a reference to estimate the meters/pixels of the video.

To track the position and orientation of SEAMOUR, the GoPro footage was processed and used for tracking using the MATLAB Image Processing Toolbox and Computer Vision Toolbox (Figure 11B). Each of the four strokes underwent three separate trials to capture the variance of the real-world system. The footage was then post processed to remove lens distortion and make lighting and color corrections. All videos were trimmed to the same duration to ensure consistency and then processed using the MATLAB Image Processing Toolbox and Computer Vision Toolbox to track the position and orientation of the robot at each frame through each trial (Figure 11B). The displacement, velocities, and pitch angle of the system were tracked for all the trials for each of the four strokes tested. The sample rate for measuring the displacement and velocities was the same as the frame rate, 24 Hz, but the sample rate for the change in pitch had to be stepped down to 3 Hz for accurate tracking by the MATLAB Computer Vision Toolbox. The initiation of each stroke was aligned, the results from each trial were averaged, and the variance between the three strokes was recorded.

## 3. Results

### 3.1. Reinforcement Learning Convergence and Performance

This work showcased the successful use of reinforcement learning to create a novel stroke for a bio-robotic sea lion platform, utilizing its numerical counterpart. Three unique learned strokes were developed after fully training a SAC reinforcement learning agent with three different initialization seeds. The highest-reward-producing gait observed within the last 10% of the 100,000 individual learning episodes was selected as the learned gait for each seed, yielding three distinct strokes. Each learning epoch required approximately 24 h to converge.

While the characteristic stroke, as previously described, includes power, paddle, and recovery phases, the RL favored strokes that eliminated the power portion and instead performed a paddle motion next to the body. During the recovery phase, both strokes A and B held the face of the foreflipper perpendicular to flow, while stroke C had a slight negative angle of attack. During the paddle portion of the stroke, strokes A and B both pitched slightly and rolled the flippers to make the flipper face flat in the direction of motion. On the other hand, stroke C did not pitch and only rolled the flipper with the face perpendicular to the direction of motion just before the paddle (Figure 12).

To evaluate how the characteristic stroke and the three learned strokes affected SEAMOUR’s performance, several metrics were used: total distance travelled in the x- and z-directions, followed by maximum and mean velocities at the end of four cycles for every stroke and the total pitch change across the four cycles. Additionally, both mean and maximum velocities, as well as pitch changes for each cycle within a stroke, are also presented and compared.

### 3.2. Strokes Applied on Numerical Model

#### 3.2.1. Translation in Simulation

All three learned strokes traveled farther in the forward direction (x-direction) compared to the characteristic stroke, but the characteristic stroke traveled the furthest in the z-direction after four consecutive cycles in simulation. The characteristic stroke translated to 1.35 m in the x-direction (Figure 13A). Stroke A translated 1.65 m in the x-direction, approximately a 22% increase in translation, and stroke B translated 1.66 m, a 23% increase in translation when compared to the characteristic stroke (Figure 13A). Stroke C translated the furthest in the x-direction, reaching 1.75 m, which was 30% further than the characteristic stroke. The characteristic stroke reached 1.04 m in the z-direction after four strokes compared to strokes A, B, and C reaching 0.18 m, 0.25 m, and −0.08 m, respectively, after the same number of strokes (Figure 13A). Stroke C resulted in the highest total translation, traveling 1.75 m (Figure 13A). The characteristic stroke had a combined translation of 1.71 m, and Strokes A and B translated 1.66 m and 1.68 m respectively (Figure 13A).

#### 3.2.2. Velocity in Simulation

In the x-direction, the learned strokes all produced higher maximum and mean velocities than the characteristic stroke over four consecutive cycles in simulation. For the characteristic stroke, the maximum velocity and mean velocity in the x-direction were 0.18 m/s and 0.11 m/s, respectively, which were the lowest values compared to all other strokes (Figure 14 and Table 1). Stroke C produced the highest velocity in the x-direction, reaching a maximum velocity of 0.27 m/s and a mean velocity of 0.15 m/s (Figure 14 and Table 1). Strokes A and B achieved maximum velocities of 0.24 m/s and 0.21 m/s, respectively, and mean velocities of 0.14 m/s and 0.14 m/s, respectively, in the forward direction (Figure 14 and Table 1). The mean velocities of strokes A, B, and C were 24−30% higher than the mean velocity produced by the characteristic stroke (Figure 14 and Table 1). The mean velocity in the x-direction increased linearly for each cycle across all strokes. For example, the mean velocity for each cycle for the characteristic stroke rose from 0.05 m/s to 0.11 m/s from cycle 1 to cycle 2 and further to 0.14 m/s and 0.15 m/s from cycle 3 to cycle 4, with an R² value of 0.90 (Figure 14 and Table 1). In contrast, the learned strokes demonstrated a linear increase with an R² value of 0.99 (Figure 14 and Table 1).

In contrast, the characteristic stroke achieved higher maximum and mean velocities in the z-direction during simulations, while the learned strokes consistently showed lower velocities in the z-direction. The characteristic stroke generated a maximum and mean velocity in the z-direction of 0.19 m/s and 0.09 m/s, respectively, after four cycles (Figure 14 and Table 1). Learned strokes A, B, and C produced much lower velocities in the z-direction with maximum velocities of 0.03 m/s, 0.05 m/s, and 0.02 m/s, respectively, and mean velocities of 0.02 m/s, 0.02 m/s, and −0.01 m/s, respectively (Figure 14 and Table 1). The characteristic stroke resulted in 4–5 times higher mean velocity than any of the learned strokes in the z-direction (Figure 14 and Table 1). The mean velocity in the z-direction increased linearly with each cycle across all strokes with an R² value of 0.99, except for Stroke C, which was trendless (Figure 14 and Table 1).

The characteristic stroke had a higher combined velocity than strokes A and B, but stroke C exhibited the highest mean and maximum combined velocity in simulation. The characteristic stroke resulted in a maximum and mean combined velocity of 0.26 m/s and 0.15 m/s, respectively, after four cycles (Figure 14 and Table 1). Learned strokes A and B resulted in lower maximum and mean combined velocities with the same mean velocities of 0.14 m/s and maximum velocities of 0.24 m/s and 0.25 m/s, respectively. Learned stroke C produced the highest maximum and mean velocity, reaching 0.27 m/s and 0.15 m/s, respectively (Figure 14 and Table 1).

#### 3.2.3. Pitch Displacement and Angular Velocity in Simulation

The characteristic stroke had the highest change in pitch (∆θ) per cycle and had the highest mean pitch angular velocity ($\bar{\dot{\theta }}$) per cycle, while all the learned strokes produced relatively low pitch changes in simulation. The characteristic stroke produced a mean ∆θ per cycle of 15.5° and a mean $\dot{\theta }$ per cycle of 5.18°/s (Figure 13B,C and Table 2). The total change of pitch angle after four cycles of the characteristic stroke was 62° in simulation (Figure 13B,C and Table 2). The learned strokes all exhibited much less change in pitch per cycle as well as low mean $\dot{\theta }$ per cycle when compared to the characteristic stroke. Strokes A, B, and C produced a ∆θ per cycle of 1.83°, 4.57°, and 2.27°, respectively (Figure 13B,C and Table 2). Learned strokes A, B, and C produced mean $\dot{\theta }$ per cycle of 0.61°/s, 1.52°/s, and 0.76°/s. For both the pitch per cycle and the mean $\dot{\theta }$ per cycle, stroke B produced about 30% of that produced by the characteristic stroke per cycle, whereas strokes A and C produced about 12–14% of the characteristic stroke’s value (Figure 13B,C and Table 2). After four cycles of each of the learned strokes, the total change in pitch angle was 18° for stroke A, 9.1° for stroke B, and 6.9° for stroke C (Figure 13B,C and Table 2).

### 3.3. Strokes Applied on Bio-Robotic Platform

#### 3.3.1. Translation on SEAMOUR

When these strokes were applied to the bio-robotic sea lion platform, the learned stroke C produced the most translation in the x-direction compared to the other learned strokes A and B, as well as the characteristic stroke at the end of four cycles. To see each of the strokes being executed on SEAMOUR refer to the Appendix A. Learned stroke C translated the most, reaching 2.52 m; stroke B translated 2.09 m; and stroke A translated the least, reaching 1.98 m (Figure 15). The characteristic stroke translated to 2.35 m (Figure 15). However, during the fourth cycle, the bio-robotic system had translated so much in the z-direction, and its nose had pitched up so significantly, that the system breached the surface (Figure 15). This caused the pitch angle to rapidly decrease, resulting in more movement in the x-direction than it would have experienced if it had not breached the surface (Figure 15).

The characteristic stroke yielded the greatest translation in the z-direction when applied to the bio-robotic system, while the learned strokes also demonstrated substantial displacement in the z-direction after four cycles. The characteristic stroke reached 1.11 m in the z-direction; this displacement was also limited by the breaching of the surface during the fourth cycle of the stroke (Figure 15). The learned strokes all translated in the opposite negative z-direction; strokes A, B, and C translated −0.5 m, −0.45 m, and −1.10 m, respectively, in the z-direction after four strokes (Figure 15). Stroke C resulted in the highest total translation, traveling 2.75 m (Figure 15). The characteristic stroke had a combined translation of 2.6 m, and strokes A and B translated 2.04 m and 2.14 m, respectively (Figure 15).

#### 3.3.2. Velocity on SEAMOUR

The characteristic stroke achieved the highest maximum velocity in the x-direction and higher mean velocities ($\bar{\dot{x}}$) than learned strokes A and B, while learned stroke C generated the second highest maximum velocity and the highest mean velocity in the x-direction after four cycles on the bio-robotic system. The characteristic stroke produced a mean velocity in the x-direction of 0.198 m/s and a maximum velocity of 0.474 m/s (Figure 16, Table 3). Specifically, it produced mean velocities of 0.18 m/s and 0.256 m/s during the first and second cycles, respectively (Figure 16, Table 3). However, the velocities reduced in the third and fourth cycles when compared to the velocity in the second cycle, with mean velocities dropping by 16% and 47% to 0.215 m/s and 0.137 m/s, respectively—the lowest among all strokes in the fourth cycle (Figure 16, Table 3). In contrast, learned stroke C, while starting with the lowest mean velocity of 0.118 m/s in the first cycle, significantly increased by 93% to 0.228 m/s in the second cycle (Figure 16, Table 3). It reached the highest mean velocity in the third cycle at 0.259 m/s and then only decreased by 5% in the fourth cycle to about 0.246 m/s (Figure 16, Table 3). Stroke C produced a mean velocity in the x-direction per cycle of 0.213 m/s and reached a peak velocity of 0.367 m/s. The learned strokes A and B produced mean velocities of 0.166 m/s and 0.176 m/s, respectively, which were about 17% and 22% lower, and maximum velocities of 0.348 m/s and 0.328 m/s, respectively, which were approximately 5% and 10% lower than stroke C (Figure 16, Table 3).

When implemented on the bio-robotic model, the characteristic stroke produced the highest velocities in the z-direction compared to all the learned strokes. Specifically, the characteristic stroke achieved a peak velocity in the z-direction of 0.339 m/s and a mean velocity of 0.121 m/s (Figure 16, Table 3). The average velocity increased with each cycle, peaking in the fourth cycle when the maximum displacement occurred due to the system breaching the surface (Figure 16 and Figure 17). The learned strokes all translated in the opposite direction (negative z-direction) and descended rather than ascended in the water column. Strokes A, B, and C produced maximum velocities in the z-direction of −0.143 m/s, −0.131 m/s, and −0.157 m/s, respectively, and mean velocities in the z-direction of −0.043 m/s, −0.040 m/s, and −0.093 m/s, respectively (Figure 16, Table 3).

The characteristic stroke produced the highest mean and maximum combined velocities of all the strokes tested after four cycles. The characteristic stroke achieved a maximum combined velocity of 0.677 m/s and a mean combined velocity of 0.245 m/s (Figure 16, Table 3). The maximum combined velocities for learned strokes A, B, and C were approximately 43–51% lower, measured at 0.360 m/s, 0.328 m/s, and 0.380 m/s, respectively (Figure 16, Table 3). Similarly, the mean combined velocities for the learned strokes A, B, and C were lower than the characteristic stroke by 23%, 18%, and 2%, measured at 0.188 m/s, 0.199 m/s, and 0.240 m/s (Figure 16, Table 3).

#### 3.3.3. Pitch Displacement and Angular Velocity on SEAMOUR

The characteristic stroke produced the highest change in pitch (∆θ) per cycle and had the highest mean pitch angular velocity ($\bar{\dot{\theta }}$) per cycle, and all the learned strokes produced relatively low pitch changes when applied to the bio-robotic system after four cycles. The characteristic stroke produced a mean ∆θ per cycle of 20.17° and a mean $\dot{\theta }$ per cycle of 7.00°/s (Figure 17 and Figure 18, Table 4). The total change of pitch angle after four cycles of the characteristic stroke was 23°, but this would have been higher if the bio-robotic system had not breached the surface of the water (Figure 17 and Figure 18, Table 4). Prior to the breach, the total change in pitch angle due to three cycles of the characteristic stroke was 52°, which was about 120% higher (Figure 17 and Figure 18, Table 4). The learned strokes all exhibited much less change in pitch per cycle: strokes A, B, and C produced a mean ∆θ per cycle of 3.35°, 9.87°, and 8.36°, respectively (Figure 17 and Figure 18, Table 4). The learned strokes produced a low mean $\dot{\theta }$ per cycle as well; learned strokes A, B, and C produced mean angular pitch velocity per cycle of 1.22°/s, 1.71°/s, and 2.92°/s per cycle (Figure 17 and Figure 18, Table 4). After four cycles of each of the learned strokes, the total change in pitch angle was 13.5° for stroke A, 14.4° for stroke B, and −16° for stroke C (Figure 17 and Figure 18, Table 4).

### 3.4. Comparative Analysis Numerical and Bio-Robotic Model

When both the characteristic stroke and the three learned strokes were implemented on the numerical and bio-robotic models, consistent trends were observed in terms of total forward distance traveled, velocities in the x-z plane, and pitch changes over four cycles. Among the learned strokes, C demonstrated superior performance compared to the characteristic stroke, achieving greater forward displacement, higher velocity in the x-direction, and minimal pitch change.

#### 3.4.1. Displacement Comparison

The general directions and relative magnitudes of the system’s translations in the x- and z-directions remained consistent across four cycles of the four different stroke types. However, strokes consistently produced greater translation when applied to the bio-robotic system compared to the numerical model. Notably, learned stroke C achieved the greatest forward displacement in both the bio-robotic system and the numerical model, translating an additional 0.52 m in the bio-robotic setup (Figure 13A and Figure 15). Although strokes A and B also traveled further in the bio-robotic system than in the numerical model, their forward displacements did not exceed that of the characteristic stroke (Figure 13A and Figure 15). Specifically, strokes A and B resulted in displacements of approximately 1.6 m in simulation versus 2.2 m in real-world free-swimming tests, which was about 38% higher (Figure 13A and Figure 15). In contrast, the characteristic stroke in simulation predicted a forward displacement of 1.3 m, significantly less than the 2.4 m observed in the bio-robotic test (Figure 13A and Figure 15).

In terms of z-direction displacements, the learned strokes consistently showed lower values than the characteristic stroke across both the numerical model and the experimental platform. Notably, there was a bias toward negative displacements when the strokes were applied to the bio-robotic system (Figure 13A and Figure 15). The characteristic stroke achieved an additional 0.06 m displacement in the z-direction on the free-swimming system, a difference that would likely have been greater had the fourth cycle not breached the surface of the pool (Figure 13A and Figure 15). In contrast, strokes A and B, when applied to the bio-robotic system, moved in the negative z-direction, recording displacements of −0.5 m and −0.45 m, respectively, compared to 0.18 m and 0.25 m in the simulation (Figure 13A and Figure 15). Stroke C exhibited the most significant disparity between simulation and real-world application, translating −0.08 m in simulation versus −1.1 m with the bio-robotic system in the z-direction (Figure 13A and Figure 15).

#### 3.4.2. Velocity Comparison

The velocities were higher in both directions for all strokes when applied to the bio-robotic system compared to the numerical model. For example, the mean velocity per cycle in the x-direction for the characteristic stroke and the learned strokes A, B, and C were 75%, 21%, 26%, and 46% higher, respectively, when applied to the bio-robotic system (Table 1 and Table 2). Similarly, the mean velocities per cycle in the z-direction for each of the learned strokes all increased in magnitude when applied to the bio-robotic system: increasing 12% for the characteristic stroke and 173%, 76%, and 1230% for strokes A, B, and C, respectively (Table 1 and Table 2). In both the numerical simulation and the experimental platform, the learned stroke C produced the highest velocities in the x-direction for most of the stroke duration (Table 1 and Table 2). Similarly, the characteristic stroke always produced the highest velocities in the z-direction in both bio-robotic system and numerical simulations (Table 1 and Table 2). In addition, the characteristic stroke exhibited higher total velocities than the learned strokes in both the numerical model and the bio-robotic system for most of the four-cycle duration (Table 1 and Table 2).

#### 3.4.3. Pitch Displacement and Angular Velocity Comparison

Both numerical simulations and bio-robotic free-swimming tests demonstrated that the characteristic stroke led to the largest pitch changes over four cycles. Specifically, each cycle of the characteristic stroke increased the model’s pitch by an average of 15° in the simulation, closely matched by a 20° increase per cycle in the bio-robotic system (Table 3 and Table 4). The peak pitch reached 62° in the simulation, whereas it was 52° in the bio-robotic model, which was about a 16% decrease (Table 3 and Table 4). Notably, the pitch change in the bio-robotic system began to decrease significantly towards the end of the stroke due to a breach during the fourth cycle (Table 3 and Table 4). In comparison, the learned strokes exhibited greater pitch changes on average in the bio-robotic system than in the numerical simulations: strokes A, B, and C resulted in mean pitch changes per cycle of 1.8°, 4.6°, and 2.3° in simulation, versus 3.4°, 9.9°, and 8.4°, respectively, in the bio-robotic tests, which were approximately 88%, 115%, and 265% higher (Table 3 and Table 4).

## 4. Discussion and Conclusions

Reinforcement learning developed a novel straight-swimming sea lion foreflipper stroke that outperformed the characteristic stroke by effectively generating an efficient swimming gait in both simulations and real-world tests. The objective was to maximize forward velocity and translation while minimizing out-of-plane motion and orientation changes. All three learned strokes achieved these goals in simulations and on the robotic platform.

Learned stroke C significantly outperformed the characteristic stroke across all primary metrics: it traveled further and faster in the x-direction, exhibited a lower change in pitch, and showed minimal translation in the z-direction in both simulation and real-world tests. In simulation, stroke C not only traveled the furthest, reaching 1.75 m—which was 30% further than the characteristic stroke—but also achieved the highest maximum velocity of 0.27 m/s and a mean velocity of 0.15 m/s. Additionally, pitching per cycle was reduced by 85% compared to the characteristic stroke. When implemented on the robotic system, the characteristic stroke translated 2.35 m in the x-direction, yet stroke C still outperformed it by traveling 2.52 m over four cycles. Although the characteristic stroke achieved the highest maximum velocity in the x-direction, stroke C was close behind with a mean velocity per cycle of 0.213 m/s and a peak velocity of 0.367 m/s. Notably, stroke C resulted in significantly less pitching compared to the characteristic stroke. Correcting for the fourth cycle of the characteristic stroke that breached the surface, the total change in pitch angle was 145% greater than that of stroke C. Overall, reinforcement learning successfully developed a stroke that is markedly more effective for straight swimming in the bio-robotic system.

In simulation, learned strokes A and B showed superior performance compared to the characteristic stroke, achieving greater translation along the x-axis, reduced movement in the z-axis, and minimized pitch, despite having lower overall velocities. However, these benefits did not carry over to real-world application on the robot, highlighting a discrepancy between simulated outcomes and actual performance. In both strokes A and B, the foreflippers’ faces were oriented forward into the flow, positioned perpendicular to the direction of movement. In the real-world robot, inherent asymmetries such as motor misalignment and chord-wise flipper compliance produced a significant roll displacement in the robot, which may have adversely affected their translation performance.

While the characteristic stroke is effective for the biological sea lion, strokes developed through reinforcement learning, tailored to the specific dynamics of the robotic system for a defined task, yield better outcomes. In experiments, the biologically derived characteristic stroke induced severe pitching in both simulation and real-world tests. In contrast, the learned strokes, designed using the bio-robotic system as a training environment, did not exhibit such drastic changes in pitch. The forces producing pitch moments during the characteristic sea lion stroke will be experimentally evaluated in an upcoming paper.

This work establishes a foundation for more advanced applications of reinforcement learning in bio-robotic propulsion control. While the biological sea lion uses its foreflippers, full body, and hind flippers for agile maneuvering, this study did not include these additional control surfaces. Incorporating them could enhance the performance of the characteristic stroke and influence the outcomes of strokes learned through reinforcement learning. Future research should also examine the direct application of reinforcement learning in real-time robotic operation. Moving beyond an open-loop kinematic approach, implementing a traditional closed-loop system that adapts based on sensory feedback from the environment could prove beneficial. This would require equipping SEAMOUR with enhanced proprioceptive sensors and hardware upgrades to ensure greater robustness.

Achieving proper alignment between the center of gravity and the center of buoyancy for both the robot and the simulation proved highly challenging, impacting the stability and performance of the robotic system. The robot operated as an open system, allowing water to ingress and egress, which made it difficult to predict changes in the center of gravity and buoyancy during operation. This unpredictable water movement could contribute to discrepancies in pitch angle between the simulation and the physical system’s gait performance, as observed in the downward translation trend of the learned strokes when applied to the bio-robotic system. Additionally, dynamic interactions between the water and the robot’s internal structures complicate these predictions, underscoring the need for more sophisticated modeling and control techniques. Future efforts should focus on calibrating the weight distribution and internal dynamics of the physical system to better align with simulation predictions and correct the slight negative buoyancy observed during experimentation. Exploring different initial conditions, such as varying starting orientations, velocities, or operating over longer time scales, along with testing from non-static starts to better mimic the natural propulsive clap of a sea lion, are promising avenues for enhancing system alignment and performance.

The reinforcement learning approach, while promising, exhibited several areas where the learning structure could be refined to improve the resulting gaits. One potential area for enhancement is the reward function. Each of the factors considered in the reward function (forward velocity, heading, angular velocity, and distance traveled forward) were equally weighted in these experiments. Using an optimization technique such as Bayesian optimization or a genetic algorithm to refine the reward function weights could lead to improved gait outcomes. Weighting the distinct factors of the reward function will influence the resulting movements; when the weights are equal, it is possible that some terms are overrepresented in their impact on the final behaviors. Another potential area for improvement is the lack of an efficiency metric within the reward function, affecting gait performance. The current setup may also benefit from structural changes, such as increasing the number of steps per training episode. As mentioned previously, adding additional steps to the learning episodes and incorporating a closed-loop framework with access to additional degrees of freedom could produce more robust reinforcement learning-developed swimming controllers. Moreover, although not within the scope of the present study, other gait development methods like genetic algorithms, various alternate types of reinforcement learning agents such as deep deterministic policy gradients (DDPG) or proximal policy optimization (PPO), and bio-inspired techniques like central pattern generators could be explored.

Reinforcement learning, when applied to complementary bio-robotic and numerical models of a California sea lion, produced a novel swimming gait that outperformed the biologically derived stroke on all relevant metrics, including forward displacement, forward velocity, and minimal pitch displacement. Gaits could further be enhanced with improved alignment of system dynamics of both the numerical model and the bio-robotic platform. Further tuning the reward function holds the potential for both improved gait performance and the adaptation of diverse gaits for specific tasks such as a slow-moving gait or maintaining a constant velocity. The successful implementation and performance of the learned stroke on the bio-robotic system using this approach highlights the potential to develop additional swimming gaits tailored to various locomotion tasks.

## Figures and Tables

**Figure 1 biomimetics-09-00522-f001:**
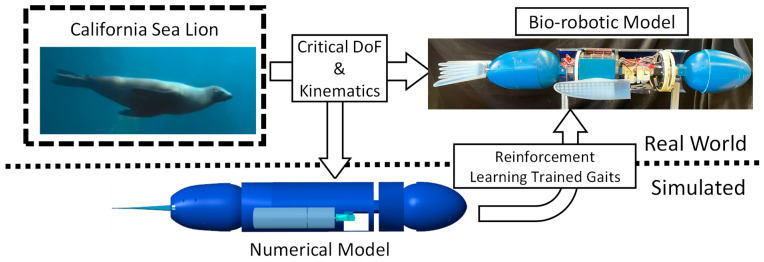
Bio-robotic and Numerical Model of the California Sea lion for Reinforcement Learning. Using the California sea lion as a model, complementary numerical and bio-robotic models were constructed utilizing the observed important degrees of freedom present in the animal. The propulsive kinematics were replicated, and reinforcement learning was applied in simulation to further modify the kinematics for direct use on the bio-robotic system.

**Figure 2 biomimetics-09-00522-f002:**
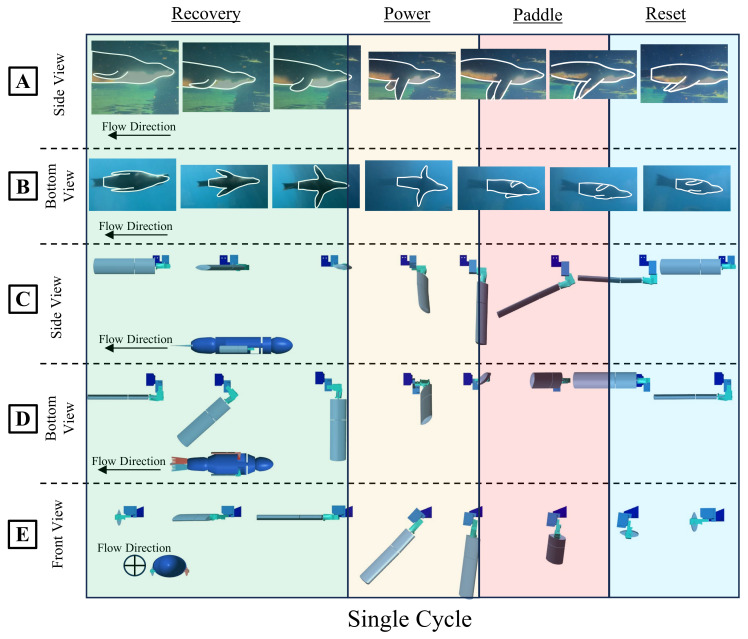
The Sea Lion Characteristic Stroke Flipper Frame Kinematics. (**A**) Side view and (**B**) bottom view of the sea lion executing its characteristic foreflipper stroke. (**C**) Side view, (**D**) bottom view, and (**E**) front view of modeled kinematics executing the characteristic foreflipper stroke in simulation.

**Figure 3 biomimetics-09-00522-f003:**
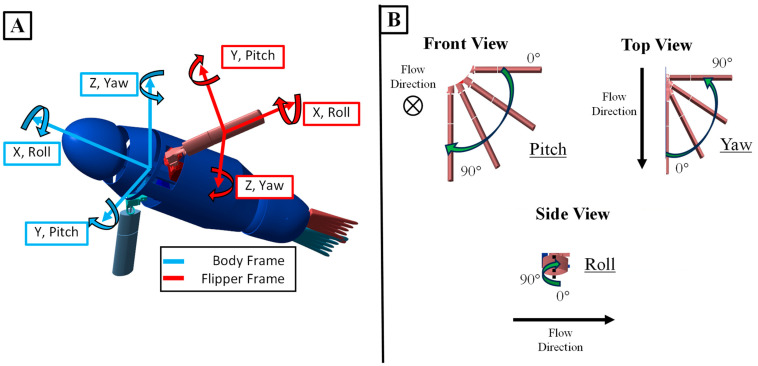
Flipper and Body Frame of Sea Lion Model. (**A**) The flipper and body frames of the sea lion model. (**B**) Specific orientation of the flipper frame.

**Figure 4 biomimetics-09-00522-f004:**
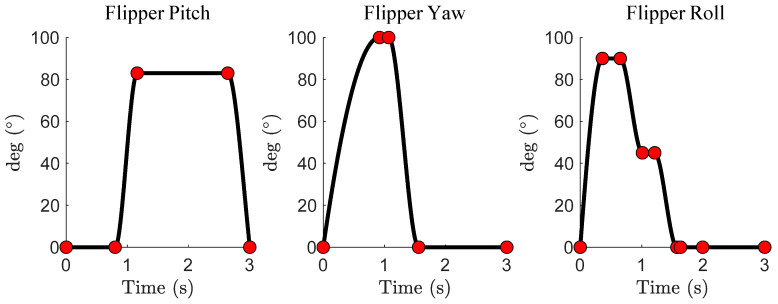
Spline Model of the Characteristic Sea Lion Stroke. The pitch, yaw, and roll angles of the characteristic sea lion propulsive stroke. The red dots are the control points that the piecewise cubic Hermite interpolating polynomial (PCHIP) is fit to.

**Figure 5 biomimetics-09-00522-f005:**
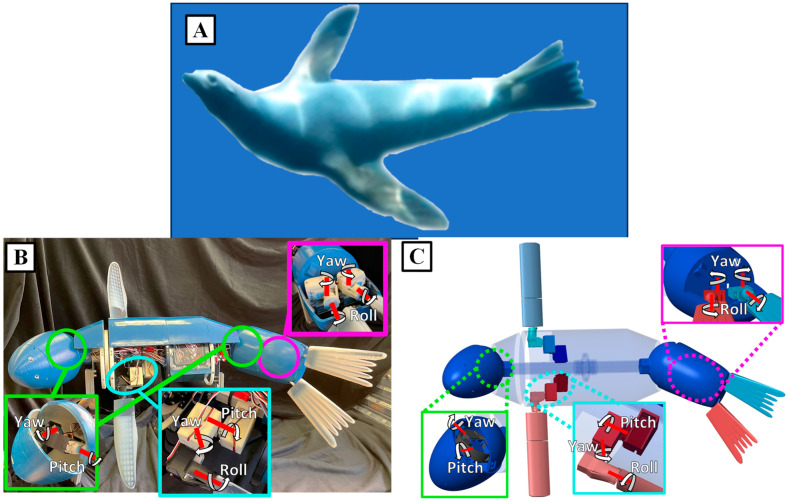
The Bio-robotic and Numerical Model of the California Sea Lion. (**A**) The California sea lion making use of its flippers and body to execute a maneuver. (**B**) The bio-robotic sea lion model with all the degrees of freedom selected to model the motions of the sea lion. (**C**) The numeric model representation of the bio-robotic model.

**Figure 6 biomimetics-09-00522-f006:**
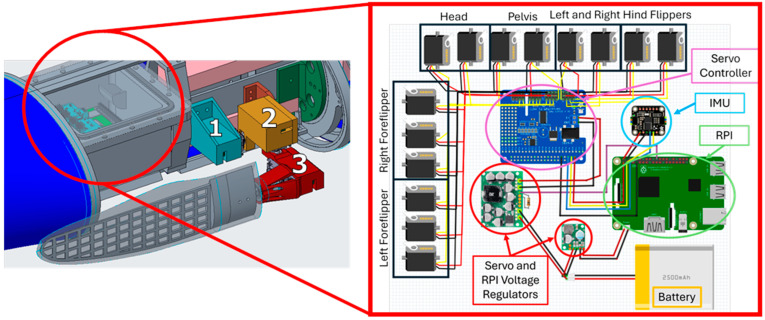
Power, Control, and Actuation of SEAMOUR: (**Left**) Power and control layout housed onboard SEAMOUR and servo motor assembly for the foreflipper (1) pitch, (2) yaw, and (3) roll. (**Right**) Power and control layout including onboard Raspberry Pi, onboard IMU, and all actuators.

**Figure 7 biomimetics-09-00522-f007:**
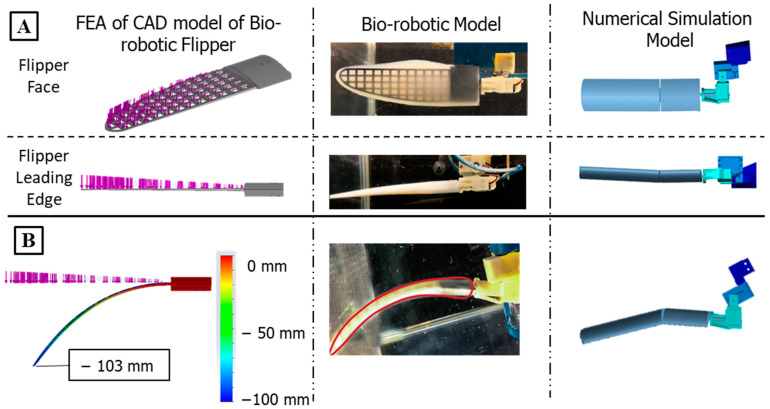
Flipper Design and Development For Numerical Model and Bio-robotic Model: (**A**) Flipper face and leading edge for the bio-robotic model and numerical model and how the FEA model distributed force along the face of the flipper. (**B**) The bending due to a 5N nonuniformly distributed load in FEA compared to the bending of the bio-robotic flipper and numerical model flipper during a propulsive stroke.

**Figure 8 biomimetics-09-00522-f008:**
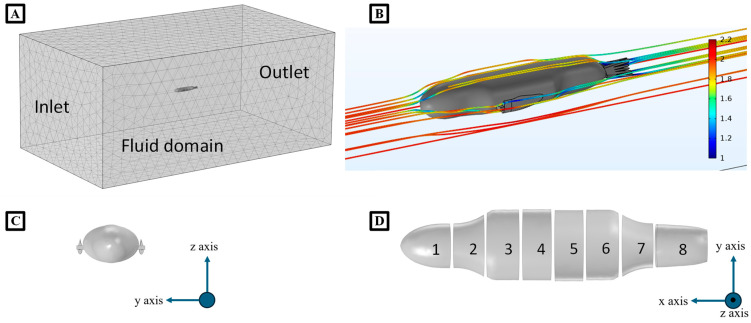
Drag Simulations of the Sea Lion Numerical Model. (**A**) CFD fluid domain. (**B**) CFD simulation to determine the coefficient of drag of the sea lion model. (**C**) Front face for calculating force coefficients in the x-direction. (**D**) Strip theory division of the body for calculating force coefficients in the y- and z-directions with numbers along the body indicating each individual strip section.

**Figure 9 biomimetics-09-00522-f009:**
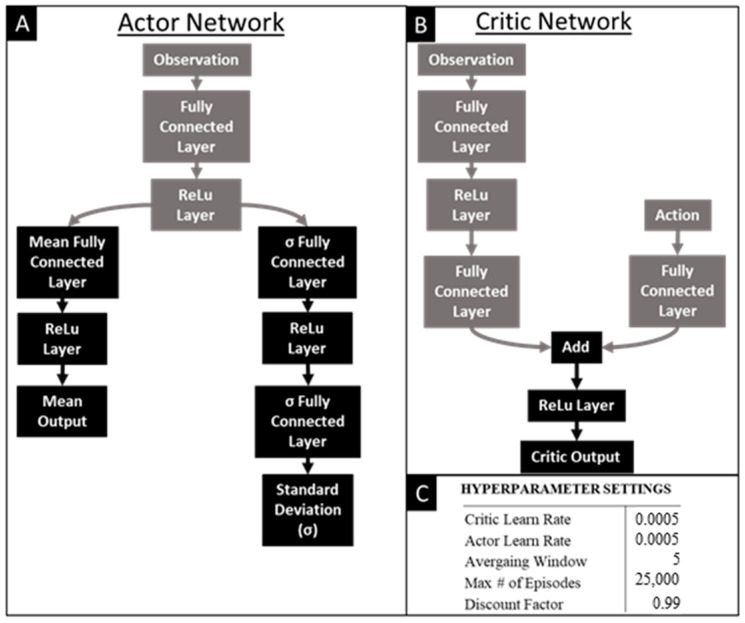
Structure of the Actor and Critic Networks and the Hyperparameters: Architecture of the (**A**) actor network and the (**B**) critic network used in the soft actor-critic (SAC) agent and the (**C**) hyperparameters.

**Figure 10 biomimetics-09-00522-f010:**
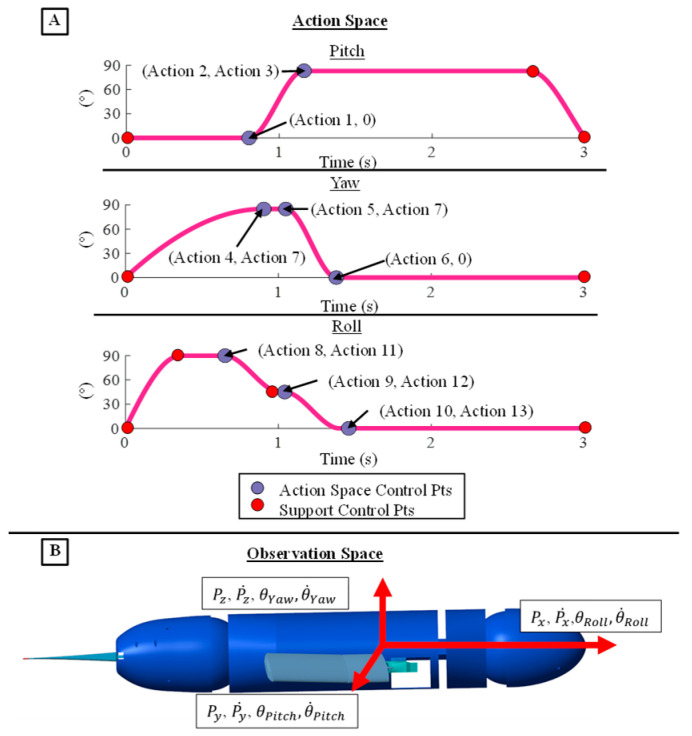
Action and Observation Space for the Reinforcement Learning Environment. (**A**) The 13 actions that control the shape, magnitude, and timing of the PCHIP spline model of the fore-flipper kinematics. (**B**) The observation space that defines the motion of the sea lion model to the reinforcement learning agent.

**Figure 11 biomimetics-09-00522-f011:**
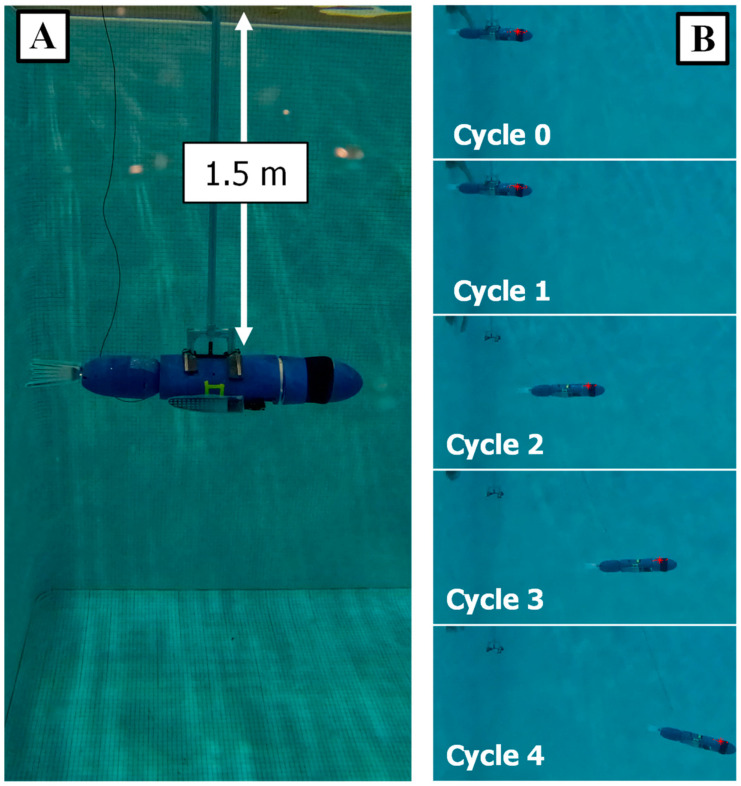
Experimental Setup: (**A**) Docking system used during experimental trials to keep consistent depth and orientation. (**B**) Five frames of tracking using the Go Pro and MATLAB Computer Vision Toolbox at the end of each of the four cycles for one of the strokes tested.

**Figure 12 biomimetics-09-00522-f012:**
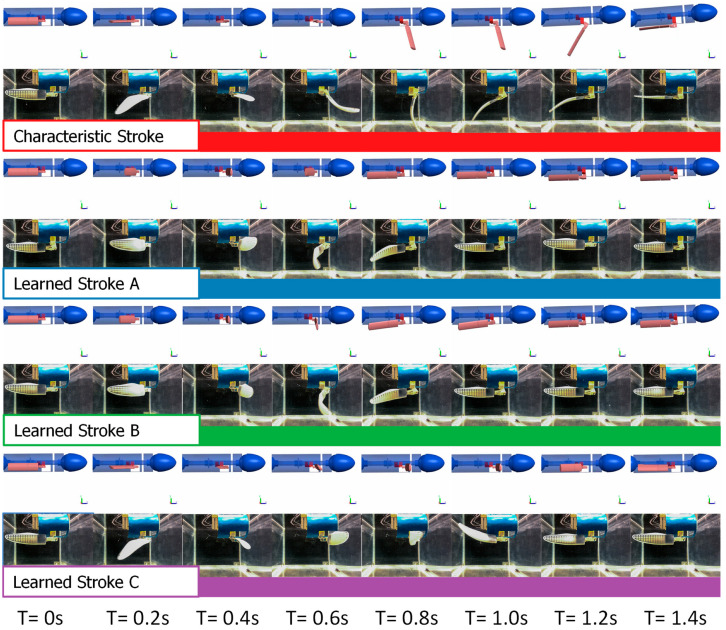
Simulation and Bio-robotic Stroke Comparison: Time sequence of four gaits tested in both the simulation and with the robotic flipper. The image shows the stroke from initiation through the stroke cycle and concludes with the return to streamline.

**Figure 13 biomimetics-09-00522-f013:**
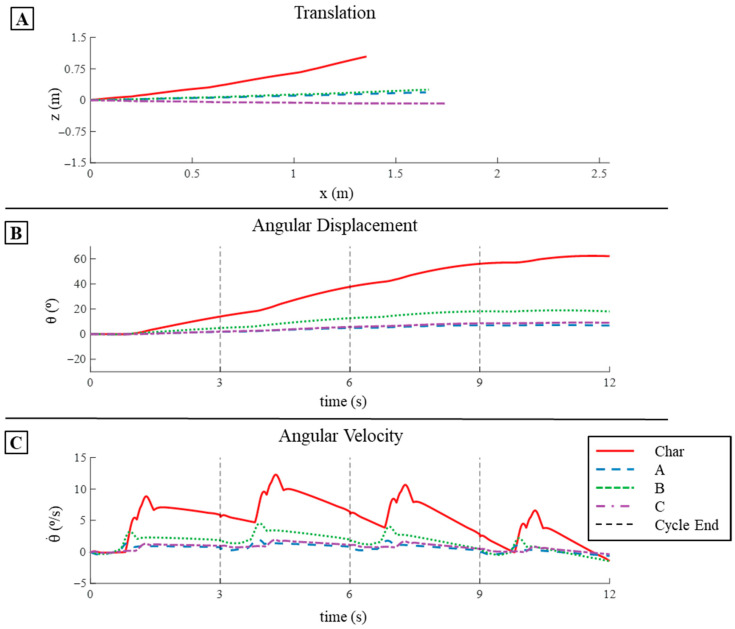
Translation and Pitch Displacement and Velocity in Simulation: (**A**) The translation in the x- and z-directions produced by the four strokes after four cycles in simulation. (**B**) The pitch displacement that occurs in simulation for the four strokes after four cycles. (**C**) The corresponding angular velocity in the pitch direction over the four cycles for each stroke.

**Figure 14 biomimetics-09-00522-f014:**
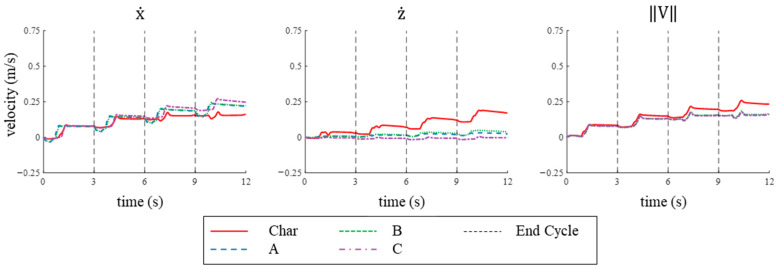
X, Z, and Total Velocities Produced in Simulation: The velocities in the x-, z-, and combined directions produced by each of the four strokes in simulation over four cycles.

**Figure 15 biomimetics-09-00522-f015:**
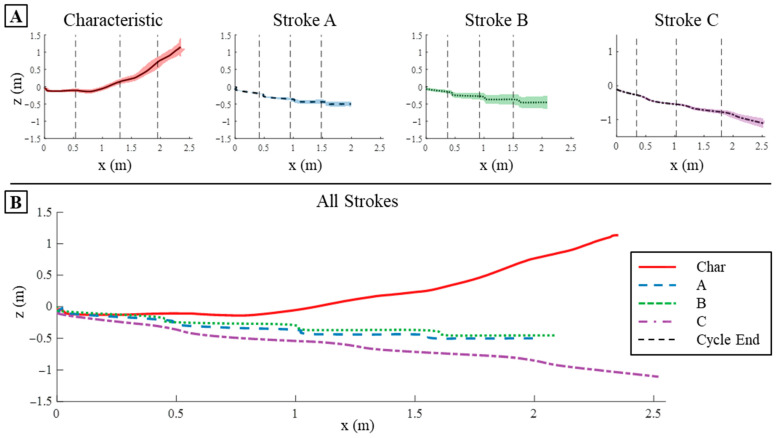
X and Z Translation on The Bio-robotic System: (**A**) The translation in x- and z-directions produced by each of the strokes individually when applied to the bio-robotic system over four cycles. The shaded region represents the variance in translation between the three trials conducted for each stroke. (**B**) The mean translation produced by each stroke over the three trials conducted.

**Figure 16 biomimetics-09-00522-f016:**
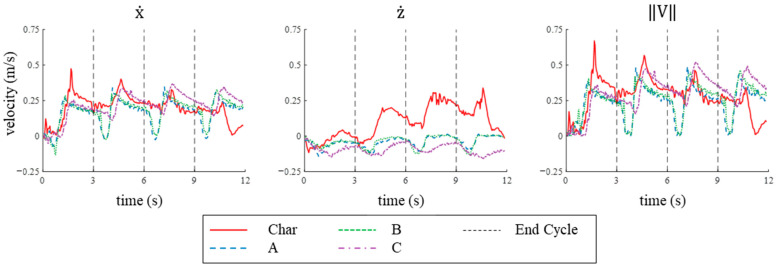
X, Z, and Total Velocities Produced on The Bio-robotic System: The velocity in the x-, z-, and combined directions produced by each of the four strokes on the free-swimming system over four cycles.

**Figure 17 biomimetics-09-00522-f017:**
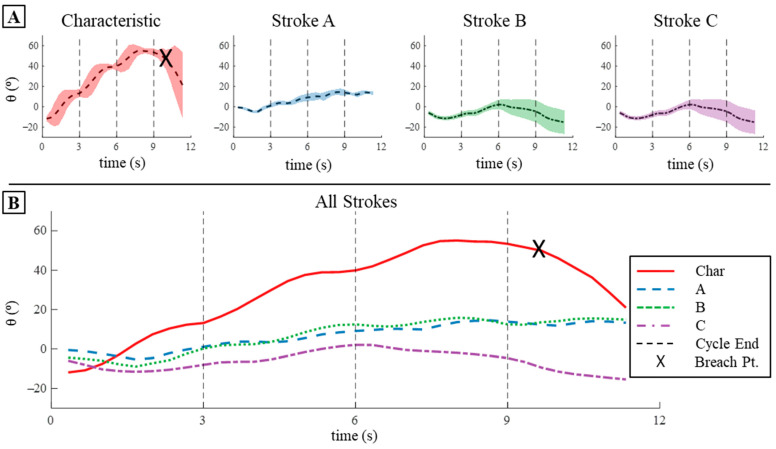
Pitch Displacement on The Bio-Robotic System: (**A**) Pitch displacement produced by each of the strokes individually when applied to the bio-robotic system over four cycles. The shaded region represents the variance in pitch between the three trials conducted for each stroke. (**B**) The mean pitch displacement produced by each stroke over the three trials conducted. Note the X that occurs during the fourth cycle of the characteristic stroke, which represents the time at which the robot ascended so far in the water column that it breached the surface.

**Figure 18 biomimetics-09-00522-f018:**
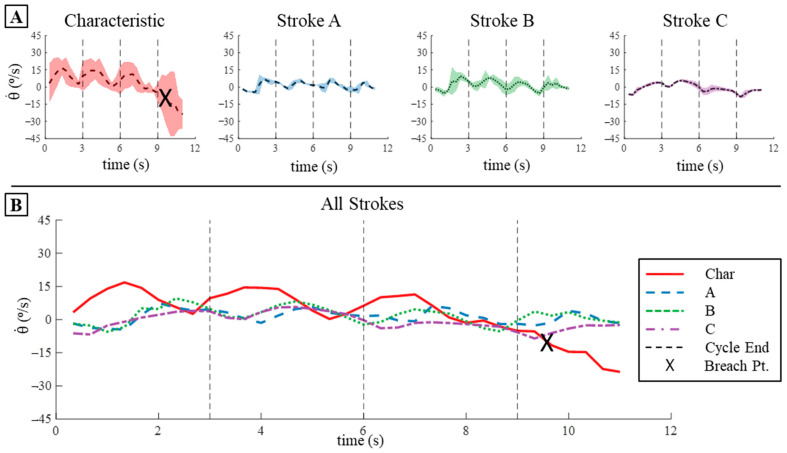
Pitch Velocity on The Bio-Robotic System: (**A**) Pitch velocity produced by each of the strokes individually when applied to the bio-robotic system over four cycles. The shaded region represents the variance in pitch velocity between the three trials conducted for each stroke. (**B**) The mean pitch velocity produced by each stroke over the three trials conducted. Note the X that occurs during the fourth cycle of the characteristic stroke, which represents the time at which the robot ascended so far in the water column that it breached the surface.

**Table 1 biomimetics-09-00522-t001:** Mean velocity in the x-, z-, and combined directions over each cycle from numerical model for each experimental foreflipper stroke.

Stroke Type	Cycle 1			Cycle 2			Cycle 3			Cycle 4			Mean		
	$\bar{\dot{x}}$	$\bar{\dot{z}}$	$\bar{\left\|V\right\|}$	$\bar{\dot{x}}$	$\bar{\dot{z}}$	$\bar{\left\|V\right\|}$	$\bar{\dot{x}}$	$\bar{\dot{z}}$	$\bar{\left\|V\right\|}$	$\bar{\dot{x}}$	$\bar{\dot{z}}$	$\bar{\left\|V\right\|}$	$\bar{\dot{x}}$	$\bar{\dot{z}}$	$\bar{\left\|V\right\|}$
Char	0.048	0.022	0.059	0.109	0.061	0.125	0.142	0.107	0.179	0.153	0.157	0.221	0.113	0.087	0.146
A	0.052	0.005	0.063	0.118	0.012	0.119	0.170	0.019	0.172	0.209	0.025	0.210	0.137	0.015	0.141
B	0.052	0.005	0.062	0.119	0.015	0.120	0.172	0.026	0.174	0.211	0.037	0.214	0.139	0.021	0.143
C	0.047	−0.004	0.165	0.121	−0.008	0.121	0.183	−0.009	0.183	0.232	−0.006	0.232	0.146	−0.007	0.147

**Table 2 biomimetics-09-00522-t002:** Change in pitch angle (θ) and mean angular velocity ($\dot{\theta }$) over each cycle from the numerical model for each experimental foreflipper stroke.

Stroke Type	Cycle 1		Cycle 2		Cycle 3		Cycle 4		Mean	
	Δθ	$\bar{\dot{\theta }}$	Δθ	$\bar{\dot{\theta }}$	Δθ	$\bar{\dot{\theta }}$	Δθ	$\bar{\dot{\theta }}$	Δθ	|$\bar{\dot{\theta }}$|
Characteristic	14.03	4.68	23.71	7.91	18.3	6.1	6.04	2.02	15.52	5.18
A	1.86	0.62	3.08	1.03	2.15	0.72	−0.25	−0.09	1.83	0.61
B	4.86	1.62	7.87	2.62	5.45	1.82	−0.1	−0.03	4.57	1.52
C	2.02	0.68	3.72	1.24	2.87	0.96	0.48	0.16	2.27	0.76

**Table 3 biomimetics-09-00522-t003:** Mean velocity in the x-, z-, and combined directions over each stroke from the bio-robotic system for each experimental foreflipper stroke.

Stroke Type	Cycle 1			Cycle 2			Cycle 3			Cycle 4			Mean		
	$\bar{\dot{x}}$	$\bar{\dot{z}}$	$\bar{\left\|V\right\|}$	$\bar{\dot{x}}$	$\bar{\dot{z}}$	$\bar{\left\|V\right\|}$	$\bar{\dot{x}}$	$\bar{\dot{z}}$	$\bar{\left\|V\right\|}$	$\bar{\dot{x}}$	$\bar{\dot{z}}$	$\bar{\left\|V\right\|}$	$\bar{\dot{x}}$	$\bar{\dot{z}}$	$\bar{\left\|V\right\|}$
Char	0.179	−0.027	0.199	0.256	0.087	0.284	0.215	0.189	0.295	0.137	0.144	0.202	0.198	0.098	0.245
A	0.135	−0.064	0.171	0.178	−0.051	0.199	0.178	−0.027	0.193	0.175	−0.022	0.19	0.166	−0.041	0.188
B	0.125	−0.047	0.161	0.184	−0.043	0.204	0.194	−0.031	0.212	0.202	−0.03	0.217	0.176	−0.037	0.199
C	0.118	−0.09	0.165	0.228	−0.091	0.251	0.259	−0.076	0.272	0.246	−0.115	0.274	0.213	−0.093	0.241

**Table 4 biomimetics-09-00522-t004:** Change in pitch angle (θ) and mean angular velocity ($\dot{\theta }$) over each stroke from the bio-robotic system for each experimental foreflipper stroke.

Stroke Type	Cycle 1		Cycle 2		Cycle 3		Cycle 4		Mean	
	Δθ	$\bar{\dot{\theta }}$	Δθ	$\bar{\dot{\theta }}$	Δθ	$\bar{\dot{\theta }}$	Δθ	$\bar{\dot{\theta }}$	|Δθ|	| $\bar{\dot{\theta }}$|
Characteristic	−1.48	1.38	27.22	10.61	18.17	6.77	−33.8	−9.24	20.17	7.00
A	−0.16	0.40	7.33	2.68	5.41	1.57	−0.5	−0.22	3.35	1.22
B	1.55	2.06	22.95	3.30	12.45	0.20	2.53	1.27	9.87	1.71
C	−9.84	−3.04	8.9	3.50	−1.96	−0.88	−12.75	−4.25	8.36	2.92

## Data Availability

The data generated during the study are available from the corresponding author on reasonable request.

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
