# Peer review of "Using Reinforcement Learning to Develop a Novel Gait for a Bio-Robotic California Sea Lion"

_biomimetics, 2024, doi:10.3390/biomimetics9090522_

Round 1

Reviewer 1 Report

Comments and Suggestions for Authors

Overall, the article lacks robust logic, does not adequately highlight innovative points, and suffers from insufficient experimental content and a lack of relevant comparisons.

Evaluation:

1. The underwater environment considered in the article is limited to hydrostatic and hydrodynamic forces, with hydrodynamic forces being simplified as drag forces only. Additional factors such as added mass force and viscous force are not considered. The simulation environment used is MATLAB-Simscape, where the forces generated by propulsion methods (pectoral and caudal fins) are completely replaced by torques, with no specific explanation provided for the substitution method.

2. Furthermore, the SAC algorithm used for simulation is overly simplistic. Additionally, please confirm using Raspberry Pi 4B for migration and deployment. The hardware requirements in current articles related to sim to real robot migration tend to be relatively high.

3. The article utilizes a motion capture system to measure the robot's speed and displacement, which seems to be offline and not typically employed for reinforcement learning methods.

Necessary Clarifications:

Detailed explanations are required for the reinforcement learning section, particularly justifying the simplification of hydrostatic and hydrodynamic forces. More extensive field tests with a wider range of categories should be conducted to demonstrate the effectiveness of the reinforcement learning training results.

Comments on the Quality of English Language

Minor editing of the English language is required

Author Response

Please see attachment for the author's response

Reviewer 2 Report

Comments and Suggestions for Authors

Dear authors! The paper presents a very interesting study dedicated to a neural network based optimizing the propulsion of a robotic sea lion. The research has been conducted on a high technical level and provides fruitful insights into design of such biomimetic swimming robots and their control systems.

Nevertheless, several remarks should be made.

1. In Figure 3, it is desired to present front view of the sea lion and bottom view to the model of the robot to show correspondence between the robot and the animal it these projections.

2. The mechanical design of foreflippers raise some questions on whether their actuator assemblies are stiff enough and do not suffer from backlashes and other mechanical issues. Please give a more detailed drawing/photo of them and give some notes on their performance in terms of stiffness, backlash, movement repeatability etc. The model of servo motors and their manufacturer should also be given.

3. In Figure 10, we see that the pitch angle of the robot was still far from 0, unlike the pitch angle in the model. You write, that this was caused by misalignment of "centers of mass and buoyancy in the real-world system". But could that be improved with additional control using pelvic flippers working as an elevator or by other active control methods? How actual sea lions cope with that taking into account notable change is their mass during seasons and even before/after a meal?

4. Why do you use rectangular shape of the flippers in the 3D model while the shape in the physical model is closer to real animal? From where the flipper shape was obtained? You write that you use uncambered airfoil shape, while the real flipper is cambered, as the work "Noninvasive 3D geometry extraction of a Sea lion foreflipper" presents. Please explain your choise. Then, you write "finite element analysis (FEA) was done to simulate the bending of the foreflipper when subjected to an estimated peak fluidic loading during a propulsive stroke". Please could you provide some results of these numerical experiments? How do they coincide with the real-world result? 

5. Please give an information on electronic control unit of the robot: its architecture, controllers used etc.

6. Some broader context should be given about biomimetic underwater propulsion research. For example, some considerations should be made about comparison of the given design with fish-like propulsion and its advantages. Some more recent works on the topic should be mentioned, e.g. "Design of Fish-like Biomorphic Propulsion" (2023).  

Reviewer 3 Report

Comments and Suggestions for Authors

The authors of the entitled manuscript "Using Reinforcement Learning to Develop a Novel Gait for a 2 Bio-robotic California Sea Lion" presented a reinforcement learning (RL) method to develop a novel sea lion foreflipper gait for a bio-robotic swimmer using a numerically modelled computational representation of the robot. The topic is very interesting to read. However, I have the following comments:

  1. The introduction section is missing to show the link between RL and CFD modelling. Therefore, the author needs to revise this section.
  2. The Geometry details should be presented before the materials and methods section. This requires to reorganize the article.
  3. The authors mentioned the use of SolidWorks FEA simulation but with no method mentioned. Show the results for the FEA in a figure.
  4. The numerical modelling information is not presented in detail. The authors mentioned the use of Simscape without describing the nature of the flow regime. These should stated in your Materials and Methods section.
  5. The CFD model was performed using COMSOL. How did the author achieve the results presented in Figure 6 A? Also, Figure 6 was presented before the text, which needs to be clarified.
  6. For the Fd calculations, show the region or faces you have selected to consider the area using one of the figures (maybe Figure 6B).
  7. Under the discussion section, the authors need to verify and quantify some of the statements. For example, "Both the numerical and physical models were sensitive to buoyancy and gravitational forces.
  8. Add a limitation section.

Minor changes:

  1. Please avoid using [] for figures.
  2. Also, Figures numbers issues. Figure 2 is presented after Figure 3.
  3. Please see the following design for prosthetic fins done by AUT. https://www.aut.ac.nz/news/stories/prosthetic-fins-for-injured-sea-turtles and https://www.aut.ac.nz/news/stories/meet-cornelia-auts-robot-turtle

Round 2

Reviewer 1 Report

Comments and Suggestions for Authors

1. In the second part of the article, the simulator simscape is used for data acquisition and intensive learning training, but the article does not introduce it in detail. Please supplement the relevant simulation environment settings, data acquisition videos and interfaces.

2. In the third part of the article, please supplement the test video corresponding to the article to further illustrate the feasibility of the algorithm. (used to evaluate the method and effect of paper presentation)

3. In the reply(R1 point3), it is not clear how raspberry pie 4b is applied to the actual reinforcement learning migration. Please supplement the specific migration application.

4. In the reply(R1 point4),GoPro does not have the ability of underwater positioning GoPro only has the ability of underwater image acquisition. Please supplement the acquisition methods of underwater specific attitude angle and position direction, and provide relevant videos.

Author Response

Please check attachment, and the video please see supplementary file

Reviewer 3 Report

Comments and Suggestions for Authors

I would like to thank the authors for answering my questions. 

Author Response

Thank you for taking the time to review our paper.